# Genomic and Demographic Characteristics of Angiosarcoma as Described in the AACR Project GENIE Registry

**DOI:** 10.3390/cancers17223663

**Published:** 2025-11-14

**Authors:** Eileen Leach, Amir Jafari, Elijah Torbenson, Beau Hsia, Abubakar Tauseef

**Affiliations:** Department of Medicine, Creighton University School of Medicine, Omaha, NE 68124, USA; amirjafari@creighton.edu (A.J.); elijahtorbenson@creighton.edu (E.T.); abubakartauseef@creighton.edu (A.T.)

**Keywords:** angiosarcoma, carcinogenesis, genomics, secondary malignancy, demographics, mutual exclusivity, co-occurrence

## Abstract

In this study, we analyzed the AACR GENIE registry to characterize genomic and demographic characteristics of angiosarcoma. Angiosarcoma carries high mortality but has low prevalence, and consequently, there are limited large-scale genomic studies. Though prior studies have identified mutations associated with angiosarcoma, this is the first study to specifically analyze angiosarcoma genomics in the AACR GENIE database. We aimed to characterize the landscape of angiosarcoma and, therefore, highlight potential therapeutic targets that may aid future management strategies. Our analysis revealed recurrent mutations in *TP53*, *KDR*, and *PIK3CA*, with several genes demonstrating co-occurrence or mutual exclusivity. There was also variation in the predominant mutations between sexes and between primary and metastatic tumors.

## 1. Introduction

Angiosarcoma is a rare, high-grade malignant tumor of endothelial cell origin, most frequently affecting the skin, soft tissue, breast, liver, or heart [1,2]. It is classified by the World Health Organization (WHO) as a malignant vascular tumor [3]. Histologically, it is characterized by irregular, anastomosing vascular channels with an infiltrative growth pattern and ill-defined borders. Angiosarcoma cells commonly express markers such as *CD31*, *CD34*, and *VEGF* [4,5]. Clinically, angiosarcoma may present as bruising, ulceration, rapidly enlarging masses, and pain or discomfort. Angiosarcomas are highly aggressive and have poor prognoses, with 5-year survival rates ranging between 12% and 35% [6].

Although it remains a rare malignancy, angiosarcoma incidence is rising worldwide. In the United States, annual cases have almost doubled, from 657 in 2001 to 1312 in 2019, with most (approximately 72%) tumors arising from cutaneous, subcutaneous, or breast tissue. Angiosarcoma can also occur in visceral sites, most commonly the liver, heart, and bone [7]. Prognosis is poor, with five-year survival rates of ~40%, which decrease to ~15% for metastatic disease [8]. An overall global prevalence has not been established, but prior studies estimate annual incidences of ~0.31/100,000 in Europe and ~0.18/100,000/year in Japan [8,9]. Overall, angiosarcoma occurs at an equal rate across sexes, though women with prior radiation for breast cancer have higher rates of breast angiosarcoma [3,10,11]. Established risk factors include radiotherapy, preexisting lymphedema, chemical exposures, hypertension, and diabetes [7,8,12].

The diagnosis of angiosarcoma can be difficult as its presenting symptoms can mimic other malignancies, such as epithelial carcinomas or anaplastic melanoma. Therefore, biopsy with histopathologic and immunohistochemical evaluation is often required for diagnosis [4]. Imaging with CT, MRI, or PET is also used, as it helps define the disease extent and to assess for potential metastatic spread [3]. Prognosis varies by stage, with patients with metastases having markedly worse 5-year survival rates compared with localized disease, at 15% and 40%, respectively [8].

Therapy is multimodal, with complete surgical resection with negative margins as the standard treatment. This is often combined with adjuvant radiotherapy to reduce the risk of recurrence [3,4]. For metastatic disease, chemotherapy with drugs such as paclitaxel, anthracyclines, and ifosfamide is commonly used [3]. Newer, targeted therapies are showing some efficacy, and other studies have touched on the combination of anti-angiogenic agents and immune checkpoint inhibitors, but these are not yet mainstays of treatment [13,14].

Angiosarcomas can be divided into primary and secondary subtypes. Primary angiosarcomas develop in patients without prior UV exposure, chemical exposure, radiation therapy, or lymphedema, while secondary angiosarcomas develop in patients who have these risk factors [3]. Molecular profiling of angiosarcoma has revealed variation in mutations depending on the subtype. Primary breast angiosarcomas frequently present with *PI3K* pathway mutations, while secondary radiation and lymphedema-associated tumors are more likely to present with *MYC* amplifications [7,15]. Studies have also shown that secondary breast angiosarcomas present with higher grade and more advanced disease [10,14]. Certain anatomical sites, such as the scalp and face, are more likely to exhibit tumors with mutations typical of UV damage and high tumor mutation burden (TMB). Interestingly, these tumors are linked with an increased response to treatment with immune checkpoint inhibitors [7,14]. In prior studies, patients with longer progression-free survival were shown to have more cytotoxic T cells, NK cells, and myeloid dendritic cells, whereas patients with tumor-associated fibroblasts fared worse [13]. Despite these insights, there is still a need for a comprehensive genomic profiling to better understand prognostic factors and to identify therapeutic targets.

Although progress has been made in angiosarcoma research, the molecular landscape across subtypes and populations is poorly defined. A better understanding of the genetic mechanisms underlying progression, resistance, and metastasis is needed to assist the improvement of diagnostic and treatment approaches. To our knowledge, this is the first study to specifically analyze angiosarcoma genomics in the American Association for Cancer Research Project Genomics Evidence Neoplasia Information Exchange (AACR GENIE) database. Through the use of the AACR GENIE database, the largest and continually expanding cancer genomics database, we aim to better characterize the landscape of angiosarcoma, refine prognostic markers, and highlight potential therapeutic targets that may aid future management strategies.

## 2. Materials and Methods

Per Creighton University’s research guidelines, this study was exempt from institutional review board approval as it utilizes de-identified data from the publicly available AACR GENIE database. The data was accessed and collected on 13 August 2025 from cBioPortal (v18.0-public) software, which includes clinical and genomic data from 2017 and later.

The AACR Project GENIE compiles genomic and clinical data from 20 leading international cancer centers into a unified database. Sequencing within GENIE is performed using platforms such as whole-genome sequencing (WGS), whole-exome sequencing (WES), and targeted gene panels spanning a range of genes. The dataset includes tumor-only samples and matched tumor–normal pairs, with the tumor-normal pairs allowing for filtering based on germline variants. For this analysis, we identified 359 angiosarcoma samples from 346 patients in the GENIE v18.0-public release. Within this cohort, 252 samples (70.2%) originated from primary tumors and 79 samples (22.0%) originated from metastatic sites, with the remainder from unspecified sites [16,17,18,19].

Though each participating center within the AACR Project GENIE consortium uses its own eligibility criteria for mutation detection and annotation, results are standardized through GENIE’s harmonization framework. This process incorporates the Genome NEXUS, using tools such as GATK for detecting variants and ANNOVAR (2 March 2025 build) for annotation. Versions of these applications may be different, thus contributing to some variability. While the GENIE database includes outcome and therapeutic response information for certain malignancies, specific treatment data are not available for angiosarcoma. Therefore, although the dataset provides harmonized genomic information, variations in pipelines, applications, and the absence of treatment data present as key limitations.

This study analyzed cases of angiosarcoma identified within the AACR GENIE dataset. Using the detailed cancer type filter, we filtered for patients with angiosarcoma, including breast and liver angiosarcoma, which are separate categories in the database. This cohort represented a subset of a larger group of soft tissue sarcomas, breast sarcomas, and hepatobiliary cancers. Our final dataset consisted of 359 tumor samples from 346 patients. In the database, samples were designated as primary when collected from the original tumor site and as metastatic when collected from a site of disease spread. To evaluate differences in genomic profiles, the frequency of individual gene mutations in primary and metastatic samples was compared via chi-squared testing.

The database provided demographic information, including sex, race, ethnicity, and age at which samples were collected, in addition to genomic data. Although different participating institutions used different sequencing panels and non-actionable genes were typically excluded, essential genes, such as *MYC*, *CRKL*, and *TP53*, were preserved across panels. Our analysis also excluded structural variants.

Copy number alterations (CNAs) in angiosarcoma samples were evaluated, with a focus specifically on amplifications and homozygous deletions, the former being the most common. The frequency of recurrent events was also calculated across the cohort. Frequencies of these recurrent CNAs were calculated in order to identify genes potentially relevant to angiosarcoma pathogenesis.

Samples with missing values were excluded from the analysis. All statistical analyses were conducted using R version 4.5.1 via RStudio (R Foundation for Statistical Computing, Boston, MA, USA), with statistical significance set at *p* < 0.05. Continuous variables were analyzed by one-sided *t*-test reported as means ± standard deviations (SD), while categorical variables were presented via frequencies and percentages. The chi-squared test was used to assess associations between categorical variables. 

To compare continuous variables, we evaluated data distribution. Normally distributed variables were compared using a two-tailed Student’s *t*-test, and non-normally distributed variables were analyzed using the Mann–Whitney *U* test. When multiple comparisons were made, the Benjamini–Hochberg false discovery rate (FDR) correction was used.

To ensure a clinically meaningful mutation set, we applied filtering criteria to the somatic mutation data. We included only nonsynonymous variants, specifically missense, nonsense, frameshift, and splice-site mutations, as these are more likely to affect protein function and contribute to oncogenesis. For a mutation to be considered, it had to exhibit a variant allele frequency (VAF) of at least 5% and be supported by a sequencing depth of 100× or greater. Due to their limited interpretive value in the context of tumor biology, we excluded mutations that were synonymous or classified as variants of unknown significance (VUS) from our analysis. The AACR GENIE database harmonized mutation annotation format (MAF) files were used for all mutation calls. The MAF files act as a centralized source of variant data, providing details on gene identifiers and associated protein changes across all participating institutions.

## 3. Results

### 3.1. Patient Demographics

This analysis consisted of 359 samples from 346 patients. Due to the limited sample size, the initial demographic analysis combined primary and metastatic tumor samples (Table 1). Samples were chiefly collected from the primary tumor site (*n* = 252, 70.2%), with 79 (22.0%) collected from metastatic sites. Overall, there were slightly more females (*n* = 194, 56.1%) than males (*n* = 141, 40.8%) in the overall angiosarcoma group, which included liver and breast angiosarcomas. However, of the 73 patients with breast angiosarcoma, most were female (*n* = 65, 90.3%), with the remaining patients being of unknown gender (*n* = 7, 9.7%). Males were predominant in the liver angiosarcoma group (*n* = 7, 70.0%), with the remaining patients being females (*n* = 3, 30.0%). Almost all patients were adults (*n* = 355, 98.9%) at the time of diagnosis, with only 4 (1.1%) patients under the age of 18 years. The peak incidence of sequencing for angiosarcoma occurred in patients between 60 and 70 years old (*n* = 99), followed by patients between 70 and 80 years old (*n* = 74). The majority of patients in our study were White (*n* = 239, 69.1%). Other racial groups analyzed included Asian (*n* = 24, 6.9%), Black (*n* = 20, 5.8%), Pacific Islander (*n* = 1, 0.3%), or other (*n* = 28, 8.1%). The race of the remaining patients was unknown/not collected (*n* = 34, 9.9%). From an ethnic standpoint, the majority of patients were non-Hispanic (*n* = 264, 76.3%), with 31 (9.0%) Hispanic patients. The remainder of patients (*n* = 51, 14.7%) had unknown ethnicity.

### 3.2. Somatic Mutations and Copy Number Alterations

The most common mutations were in *TP53* (*n* = 74, 20.6%), *KDR* (*n* = 49, 13.6%), *PIK3CA* (*n* = 38, 10.6%), *KMT2D* (*n* = 33, 9.20%), *FLT4* (*n* = 33, 9.20%), *FAT1* (*n* = 29, 8.08%), *POT1* (*n* = 27, 7,52%), *ATM* (*n* = 25, 6.96%), *ARID1A* (*n* = 23, 6.40%), *NOTCH 2* (*n* = 19, 5.29%), *NOTCH1* (*n* = 19, 5.29%), *ATRX* (*n* = 19, 5.29%), *NRAS* (*n* = 18, 5.01%), and *HRAS* (*n* = 18, 5.01%) (Table 2).

Of the samples analyzed, copy number alterations were found in 289 samples. Amplification events were the most common alterations and were seen in tumor suppressor genes, including *MYC* (*n* = 79, 27.3%), *CRKL* (*n* = 30, 10.4%), *FLT4* (*n* = 16, 5.54%), *KDR* (*n* = 14, 4.84%), *MAPK1* (*n* = 13, 4.50%), and *KIT* (*n* = 11, 3.81%). Homozygous deletions were less common than amplification events but still occurred, most commonly in *CDKN2A* (*n* = 19, 6.57%), *CDKN2B* (*n* = 19, 6.57%), and *MTAP* (*n* = 11, 3.81%) (Figure 1). Commonly, mutations in *KDR* occur in the I-set region (Figure 2). Mutations in *TP53* commonly occur in the P53 region (Figure 3).

### 3.3. Mutational Landscapes by Sex

When grouped by sex, female patients exhibited significant enrichment for *MYC* (*n* = 66 vs. *n* = 8, *p* = 2.14 × 10^−10^) and *HRAS* (*n* = 14 vs. *n* = 3, *p* = 0.0456) mutations. Males had a significant enrichment of more mutations than females. Notably, males had higher frequencies of enrichment in *POT1* (*n* = 20 vs. *n* = 6, *p* = 3.870 × 10^−4^), *NTRK2* (*n* = 8 vs. *n* = 1, *p* = 4.232 × 10^−3^) and *FAT1* (*n* = 19 vs. *n* = 10, *p* = 9.757 × 10^−3^). Mutations in *FGFR4* (*n* = 6 vs. *n* = 0, *p* = 4.578 × 10^−3^) and *PIK3C2B* (*n* = 6 vs. *n* = 0, *p* = 8.582 × 10^−3^) were exclusively found in males.

### 3.4. Mutational Landscapes by Race

White patient samples were then compared to non-White (Black, Asian, Pacific Islander) patient samples. White patients were found to have a higher frequency of mutation in three genes: *MYC* (*n* = 64 vs. *n* = 4, *p* = 0.0121), *GLI3* (*n* = 3 vs. *n* = 0, *p* = 0.0307), and *MAPK7* (*n* = 1 vs. *n* = 0, *p* = 0.0417).

Asian patient samples were compared to non-Asian (White, Black, Pacific Islander) patient samples, and there was found to be a significantly increased frequency of mutation in several genes: *CHEK1* (*n* = 2 vs. *n* = 0, *p* = 7.950 × 10^−3^), *RBM10* (*n* = 2 vs. *n* = 0, *p* = 8.197 × 10^−3^), *CBL* (*n* = 3 vs. *n* = 3, *p* = 8.580 × 10^−3^), *EXT1* (*n* = 1 vs. *n* = 0, *p* = 0.0115), *FGF3* (*n* = 2 vs. *n* = 0, *p* = 0.0116), *BRAF* (*n* = 4 vs. *n* = 8, *p* = 0.0124), and *SETD2* (*n* = 3 vs. *n* = 5, *p* = 0.0221).

Black patient samples were compared to non-Black (White, Asian, Pacific Islander) patient samples and found to have significantly higher frequency of mutation in several genes, including *PRPF40B* (*n* = 2 vs. *n* = 1, *p* = 3.484 × 10^−3^), *CREBBP* (*n* = 3 vs. *n* = 3, *p* = 5.147 × 10^−3^), *AURKA* (*n* = 2 vs. *n* = 1, *p* = 0.0135), *KDM6A* (*n* = 2 vs. *n* = 2, *p* = 0.0258), *PARP1* (*n* = 2 vs. *n* = 2, *p* = 0.0271), *STAT3* (*n* = 2 vs. *n* = 2, *p* = 0.0279), and *PPM1D* (*n* = 2 vs. *n* = 3, *p* = 0.0472).

Finally, the single Pacific Islander patient was compared to non-Pacific Islander (White, Asian, Black) patients. This patient had a significant frequency of mutations in *PMS1* (*n* = 1 vs. *n* = 1, *p* = 7.042 × 10^−3^), *BORCS8-MEF2B* (*n* = 1 vs. *n* = 2, *p* = 0.0101), *FOXA1* (*n* = 1 vs. *n* = 3, *p* = 0.0154), *CYLD* (*n* = 1 vs. *n* = 3, *p* = 0.0163), *CALR* (*n* = 1 vs. *n* = 5, *p* = 0.0241), *PIK3CB* (*n* = 1 vs. *n* = 5, *p* = 0.0305), and *ARID1B* (*n* = 1 vs. *n* = 10, *p* = 0.0391).

### 3.5. Co-Occurrence and Mutual Exclusivity

Several gene pairs, including *FAT1* and *NOTCH2* (*p* < 0.001), *TP53* and *ATRX (p* < 0.001), *NOTCH1* and *ARID1A* (*p* < 0.001), *TP53* and *FAT1* (*p* < 0.001), and *ARID1A* and *NOTCH2* (*p* = 0.002), were found to frequently co-occur. Mutual exclusivity only occurred at a significant level in 2 gene pairs: *KDR* and *FLT4* (*p* = 0.022); and *KDR* and *ATRX* (*p* = 0.028).

### 3.6. Metastatic vs. Primary Mutations

There were 242 patients with primary cases and 69 patients with metastatic cases of angiosarcoma. Eight patients had both primary and metastatic cases. In total, 252 primary and 79 metastatic samples were included in the gene analysis. *MAPK7* mutations solely occurred in primary tumors (*n* = 1 vs. *n* = 0, *p* = 0.0253). In metastatic samples, *ZFHX4* (*n* = 2 vs. *n* = 0, *p* = 0.0392), *FGFR1* (*n* = 3 vs. *n* = 1, *p* = 0.0436), *MSI2* (*n* = 2 vs. *n* = 0, *p* = 0.0437), *HIST1H1C* (*n* = 2 vs. *n* = 0, *p* = 0.0467), and *TOP1* (*n* = 2 vs. *n* = 0, *p* = 0.0497) mutations occurred at higher frequencies compared to primary tumors.

### 3.7. Breast Angiosarcoma vs. Liver Angiosarcoma

Breast angiosarcoma and liver angiosarcoma were subdivisions of the angiosarcoma category. The breast angiosarcoma group consisted of 73 samples from 72 patients, and the liver angiosarcoma group consisted of 10 samples from 10 patients. *MYC* mutations occurred significantly more frequently in breast (*n* = 35 vs. *n* = 0, *p* = 1.375e-3) than liver angiosarcomas, while *RPTOR* (*n* = 2 vs. *n* = 1, *p* = 0.0408) and *TP53* (*n* = 4 vs. *n* = 9, *p* = 0.0456) mutations occurred more frequently in liver than breast angiosarcomas.

## 4. Discussion

In this study, we described the genomic landscape of angiosarcoma, using data from the AACR GENIE database. By analyzing 359 samples from 346 patients, we identified recurrent alterations in key oncogenic drivers and observed notable differences between primary and metastatic tumors. Our findings revealed notable variations across sex and race, identified subtype-specific mutational patterns, and underscored the frequent involvement of cancer-related pathways such as *TP53*, *MAPK*, and *PI3K* signaling.

Overall, rates of angiosarcoma have been increasing in the United States over the past 20 years [7]. Several factors may contribute to this trend. Firstly, the population of the US is aging, with a 38% increase in people over 65 years from 2010 to 2020 [20]. Our study found the highest incidence of angiosarcoma in patients aged 60–70 years (*n* = 99), followed by those aged 70–80 years (*n* = 74). This is consistent with prior studies that have shown angiosarcoma incidence increases with age [21]. In the largest cross-sectional study to date, 50% of cases between 2000 and 2021 occurred in patients aged 59 to 80 years with a median age of 71 years, consistent with the AACR GENIE database [7]. Additionally, angiosarcoma has been associated with prior radiation exposure [3,7]. Given that older patients are more likely to have a history of malignancy treated with radiation, they may be at greater risk for developing radiation-associated angiosarcoma.

Prior studies have compared incidences of angiosarcoma across sex, though the distribution has varied considerably depending on the tumor subtype and location. Overall rates of angiosarcoma appear to be equal across sex [3,11], though in our analysis, more females than males were represented (*n* = 194 vs. *n* = 141, respectively). However, it is important to note that this difference included a subset of patients (*n* = 73) with breast angiosarcoma, with a majority of them being female (*n* = 65). In contrast, 7 of the 10 patients with liver angiosarcoma were male. Excluding these two subtypes, the incidence of angiosarcoma was approximately equal across sexes (females *n* = 128; males *n* = 134). This is consistent with prior studies reporting no strong overall sex bias [3,11].

AACR GENIE lacks data on the location of angiosarcoma outside of the breast and liver, so we could not assess sex distribution for other primary locations of angiosarcoma. Some prior studies have found that angiosarcoma of the head and neck is more common in males, while thoracic and breast angiosarcomas are more common in females [7,9,11,22]. Though we are unable to comment on the sex distribution of head, neck, and thoracic angiosarcoma, we did find that a majority of breast angiosarcomas occurred in women (*n* = 65, 90.3%). This is likely due to higher rates of radiation-associated angiosarcoma after breast cancer, which is more common in females [3,7,11,22]. Liver angiosarcoma has been associated with exposures to chemicals, including steroids, vinyl chloride, thorium dioxide, arsenic, and radium [3]. Our data corroborates prior studies that have found higher rates of liver angiosarcomas in men [23]. Increased occupational exposure to such chemicals may explain these higher rates among men.

Racial differences in angiosarcoma incidence were also evident, though there is some limitation in the demographic data available. The most common race of angiosarcoma patients in our dataset was White (*n* = 239, 69.1%), consistent with prior analyses in the U.S., which reported >80% of angiosarcoma patients being White [7,24,25]. In our analysis of the AACR database, 69.1% of patients were White, which is lower than in prior studies. However, this is still higher than expected given that the U.S. population is approximately 57% White [26].

The second most common race among patients in our study was Asian. Limited studies exist on rates of angiosarcoma among Asian patients in the U.S., with existing studies primarily focusing on the comparison of White and Black patients; however, Wagner et al. report that only 3% of angiosarcoma patients in their analysis were Asian. Our study had Asian patients accounting for 6.9% of cases, notably higher than the 3% reported by Wagner et al., though approximately equivalent to the 6.0% of the U.S. population that is Asian [7]. Wali et al. found that Asians and Pacific Islanders make up 17% of hepatic angiosarcomas [23]. This could explain the higher-than-expected rates of angiosarcoma in Asians; however, further conclusions are speculative, due to the limited numbers of specifically designated hepatic angiosarcomas in our dataset. Takemori et al. reported lower rates of angiosarcoma in Japan and hypothesized that, in addition to multifactorial differences across countries, there may also be a racial component contributing to this difference [9].

Black patients comprised the third-largest racial group in our study, at 5.8%, which is lower than expected based on their 12% representation in the U.S. population [26]. Most prior studies, including racial comparisons, have primarily focused on Black vs. White patients and found rates of angiosarcoma ranging from 7 to 9% in Black patients [7,24]. Past research has found that Black patients have higher rates of hepatic and cardiac angiosarcomas compared to cutaneous angiosarcomas [23,25]. It is possible that our study included fewer patients with cardiac and liver angiosarcomas than other studies, which could explain the lower-than-expected rates among Black patients. Additionally, it is possible that there are yet to be discovered genetic components making Black patients less disposed to angiosarcoma compared to White patients.

Finally, Hispanic patients comprised 9.0% of our dataset, which is underrepresented compared to their 18% proportion of the U.S. population. Pacific Islanders were present at 0.3%, close to their 1% representation [26]. Wali et al. reported relatively higher rates of hepatic angiosarcoma in both populations, but few prior studies have focused on these ethnic groups specifically [23]. Our hepatic angiosarcoma dataset was small, so we cannot comment on whether Hispanic and Pacific Islander patients had higher rates. It is possible that there are underlying genetic components contributing to lower rates of angiosarcoma among these groups, but this is speculative. The AACR GENIE dataset allows race to be classified as unknown, not collected, or other, which may have skewed percentages. Additionally, the race and ethnicity of patients were separated and listed under different categories, with ethnicity being categorized as Non-Hispanic (*n* = 264), Unknown/Not Collected (*n* = 51), and Hispanic (*n* = 31), which is another possible explanation for lower or higher than expected rates of angiosarcoma among certain races.

It is challenging to determine associations between angiosarcoma and specific mutations, as it is a rare and highly heterogeneous tumor [27]. In our analysis of the AACR GENIE database, the most common mutations were in *TP53 (n* = 74, 20.6%), *KDR* (*n* = 49, 13.6%), *PIK3CA* (*n* = 38, 10.6%), *KMT2D* (*n* = 33, 9.20%), *FLT4* (*n* = 33, 9.20%), *FAT1* (*n* = 29, 8.08%), *POT1* (*n* = 27, 7.52%), *ATM* (*n* = 25, 6.96%), *ARID1A* (*n* = 23, 6.40%), *NOTCH2* (*n* = 19, 5.29%), *NOTCH1* (*n* = 19, 5.29%), *ATRX* (*n* = 19, 5.29%), *NRAS* (*n* = 18, 5.01%), and *HRAS* (*n* = 18, 5.01%). Prior studies have found similar mutation patterns, particularly in individual genes, including *TP53*, *KDR*, *PIK3CA*, *ATRX*, *POT1*, *FLT4*, *ARID1A*, and *ATM*, and the *RAS* family of genes [27].

*TP53*, a tumor suppressor gene that regulates the cell cycle and apoptosis, is one of the most commonly mutated genes across cancer types and was also the most frequent gene mutation in the AACR GENIE database. This finding is consistent with prior studies, with Benton et al. reporting *TP53* mutations as the most common. Additionally, they found *TP53* to be especially enriched in head and neck angiosarcoma [27]. The AACR GENIE database does not provide data on specific tumor location, so we are unable to comment on whether *TP53* mutations were associated with head and neck angiosarcoma. AACR GENIE does provide data on angiosarcoma of the breast and liver, and in our analysis, *TP53* mutations were significantly more frequent in liver angiosarcoma compared to breast angiosarcoma (*n* = 9 vs. *n* = 4; *p* = 0.0456).

Prior studies have found that KDR, which encodes for a *VEGFR2* receptor, is another frequently mutated gene in angiosarcoma [27]. In our analysis of the AACR GENIE database, *KDR* was the second most commonly mutated gene, while Benton et al. reported it as the third most commonly mutated gene. *KDR* mutations have been associated with cardiac and primary breast angiosarcomas, and in our cohort, *KDR* was present in 17 (23.3%) of breast angiosarcoma samples, supporting its role in this subtype [27].

*FLT4* is another well-studied gene mutation in angiosarcoma, associated with angiosarcomas of the head and neck, as well as secondary angiosarcomas due to radiation. Similarly to *KDR*, *FLT4* codes for another *VEGF* receptor known as *FLT4*. In cases of secondary angiosarcoma, *FLT4* amplification typically co-occurs with *MYC* [27]. In our study, *FLT4* mutations were prevalent among breast angiosarcomas (*n* = 11, 15.1%) and did co-occur with *MYC* amplifications (*p* < 0.001), consistent with the theory that many breast angiosarcomas develop secondary to radiation for primary breast cancer.

*FAT1*, a gene with tumor suppressor functions in other cancers, was among the commonly mutated genes in our analysis. Some studies propose that *FAT1* inhibits mitochondrial respiration, a process involved in vascular muscle proliferation, but its role in angiosarcoma remains to be clarified [28]. While van Ravensteijn et al. classified it as a variant of unknown significance, its high mutation frequency (8.08%) and enrichment in male patients (*n* = 19 vs. *n* = 10, *p* = 0.0098) suggest it may play a role in the development of angiosarcoma.

Mutations in the *RAS* pathway are common in various cancers and were also seen in our cohort. The *RAS* pathway is involved in regulating cell proliferation and differentiation, and when mutated, can contribute to tumorigenesis [29]. In our study, *NRAS* and *HRAS* mutations were present, consistent with prior studies [3,27,30]. Although *KRAS* mutations have been linked to hepatic angiosarcoma in other studies, we did not find a significant difference in *KRAS* mutation frequency between hepatic and breast angiosarcomas (*p* = 1.00) [7].

Among primary breast angiosarcomas, mutations in *PIK3CA* and *KMT2D* are common, with *PIK3CA* being associated with a worse prognosis in prior studies [15,27]. *PIK3CA* encodes a subunit of phosphatidylinositol 3-kinase (PI3K), which is essential to cell survival, while *KMT2D*, which encodes a histone modulator, has both tumor suppressor and tumor promoter properties [27,31]. To better contextualize the breast angiosarcoma subset in our cohort, we compared the alterations identified in the GENIE data with those reported in prior genomic series. Among the 73 breast angiosarcoma samples (representing 72 patients), we found recurrent MYC amplification (35/73, 47.9%), as well as alterations in KDR/VEGFR2 (17/73, 23.3%), FLT4/VEGFR3 (11/73, 15.1%), PIK3CA (9/73, 12.3%), KMT2D (10/73, 13.7%), and TP53 (4/73, 5.5%) (Table 3). These trends are in line with previously described differences between primary and secondary breast angiosarcoma, in which PI3K pathway and chromatin-modifying gene alterations (such as PIK3CA and KMT2D) are more common in primary tumors [15,27]. MYC amplification, often accompanied by FLT4, has been repeatedly documented in radiation-associated disease [27,30]. The presence of KDR and FLT4, both VEGF-pathway alterations, within this subgroup underscores the angiogenic signaling dependence that has been described in earlier studies. This may inform future exploration of biomarker-driven anti-angiogenic strategies [27].

Among head and neck angiosarcomas, *TP53*, *ATRX*, *ATM*, *ARID1A*, *POT1*, and *FLT4* have been reported as common mutations [15,27]. POT1, a gene involved in regulating telomere length, has been shown to be mutated in head and neck angiosarcoma three times more frequently than in other angiosarcomas. Cardiac angiosarcomas, though rare, have also been associated with POT1 mutations [27]. POT1 has specifically been associated with Li-Fraumeni-like syndrome [27,32]. Case reports have also described family members with *POT1* mutations developing cardiac and breast angiosarcomas [32]. We cannot comment on whether these *POT1* mutations in our study were associated with head and neck angiosarcomas due to data limitations. It was of note that in our cohort, *POT1* was significantly more common in males than in females (*n* = 20 and *n* = 6, respectively, *p* = 0.0003870). *ATRX* encodes for a chromatin-remodeling protein [33]. It commonly exhibits loss-of-function mutations in hepatic angiosarcomas but is also associated with head and neck angiosarcomas [27,33]. *ARID1A* encodes a subunit of the *SWI/SNF* chromatin remodeling complex, which regulates DNA accessibility for transcription, replication, and repair. Loss-of-function mutations disrupt chromatin organization and gene expression and can drive oncogenesis [34]. *ATM* encodes for a serine/threonine kinase that regulates DNA repair, cell-cycle control, apoptosis, and oxidative stress responses. The loss of *ATM* impairs these functions and may lead to genomic instability, tumor development, and an increased reliance on PARP; this opens the door to targeted therapies, such as PARP inhibitors [35].

The *NOTCH* pathway is involved in tumor suppression, and loss of function in these genes can lead to the development of cancers, including angiosarcoma [33]. We identified mutations in both *NOTCH1* (*n* = 19, 5.29%) and *NOTCH2* (*n* = 19, 5.29%) in the database, consistent with prior findings of mutations in both human and animal forms of angiosarcoma [27,33]. *NOTCH1* specifically has been linked to cutaneous angiosarcomas, while *NOTCH2* has been associated with visceral angiosarcomas with poorer survival than *NOTCH1* [33].

Secondary angiosarcomas may arise due to radiation, ultraviolet exposure, or chronic lymphedema. These secondary angiosarcomas show mutations in the DNA damage response pathway compared to primary angiosarcomas [30]. For instance, cutaneous angiosarcomas of the head commonly have ultraviolet-induced mutations [7]. These secondary malignancies often present with mutations in the DNA damage pathway and display mutations in genes such as *TP53*, *ATRX*, *FLT4*, and *ATM*, all of which were common in our analysis [27,30]. Although we lacked medical histories in the AACR GENIE to confirm radiation or UV exposure, the rate of certain mutated genes suggests that a large proportion of tumors may be due to secondary angiosarcomas. This aligns with other studies hypothesizing that the rising incidence of angiosarcoma is largely due to the increase in incidence of secondary cases [7].

Interestingly, *MYC*, *PTPRB*, and *CRKL* mutations were not common in our cohort, despite being among the most frequent mutations in other studies. *MYC* mutations have been associated with secondary angiosarcomas [27,30]. In our study, *MYC* mutations occurred in only four patients (1.1%), while other mutations associated with secondary angiosarcoma (*ATM*, *ATRX*, *TP53*, *FLT4*) were significantly more common. We hypothesize that the lower-than-expected rates of *MYC*, *PTPRB*, and *CRKL* mutations in our dataset might have been due to the small number of patients. In our dataset, MYC mutations occurred significantly more frequently in the breast compared to liver angiosarcomas (*p* = 1.376 × 10^−3^). Interestingly, *MYC* in a larger analysis of multiple studies has been less commonly found to be mutated in primary breast cancer than in other types of angiosarcoma [27].

Given the genetic variance of angiosarcoma, we performed a focused analysis of genes that were most frequently altered in our cohort and with known functional or therapeutic relevance. Mutations in *TP53*, the *VEGF* signaling pathway, and *PIK3CA* were identified as key mutations in the AACR GENIE database, each reflecting important changes to oncogenic pathways. Below, we evaluate these genes in detail by comparing our findings to prior studies and discussing their biological and therapeutic significance.

*TP53* was the most commonly mutated gene in our cohort, occurring in 74 of 359 samples (20.6%). This is consistent with the findings of Espejo-Freire et al., who reported *TP53* mutations in 29% of angiosarcoma cases [15]. Benton et al. also reported *TP53* to be the most commonly mutated gene across multiple angiosarcoma subtypes, such as cutaneous and visceral [27]. Young et al. found that TP53 mutations are particularly common in head and neck angiosarcomas associated with UV light exposure [3]. *TP53* encodes a tumor suppressor that regulates the cell cycle, DNA repair, and apoptosis. Loss-of-function mutations lead to genomic instability and unchecked proliferation. In angiosarcoma, *TP53* dysfunction is associated with both primary and secondary forms, particularly those caused by environmental DNA damage, including ultraviolet radiation and radiotherapy [3,12,15,27]. *TP53* is considered an early driver in many cancers, and its presence may indicate early DNA damage accumulation [7,30]. In angiosarcomas secondary to radiation, *MYC* mutations have been well documented. While the exact timing between *TP53* and *MYC* mutations is not yet established, one may speculate that the *p53* dysfunction may lay the foundation for subsequent mutational events [9,27,36]. Several strategies are emerging with regard to the targeting of *TP53*. Nishikawa and Iwakuma describe approaches such as reactivating mutant *p53* with APR-246, readthrough therapy for nonsense mutations, or inducing degradation of missense-mutated *p53* with HSP90 inhibitors or statins. Another strategy is to cause targeted death in cells with *p53* mutations or deletions. These include *WEE1* or *CHK1/2* inhibition, as well as inhibiting downstream pathways like *YAP/TAZ*. The development of these approaches suggests therapies targeting *TP53* could become a treatment modality for angiosarcoma [37].

*KDR* (*VEGFR2*) mutations were present in 49 of 359 samples (13.6%), and *FLT4* (*VEGFR3*) mutations in 33 of 359 samples (9.2%). These findings are consistent with prior studies reporting *KDR* and *FLT4* mutations in approximately 10-15% of angiosarcomas [15,27,38]. In our analysis, *KDR* and *FLT4* mutations were mutually exclusive (*p* = 0.022), which may suggest redundancy within the *VEGF* signaling pathway. The vascular endothelial growth factor pathway (*VEGF*) plays a role in regulating endothelial cell survival and growth and is a central regulator of angiogenesis [27]. *KDR* encodes *VEGFR2*, a key mediator of angiogenesis, while *FLT4* encodes *VEGFR3*, which regulates lymph angiogenesis and vascular remodeling [39]. Mutations in these genes are believed to drive ligand-independent receptor activation, leading to continuous angiogenic signaling and abnormal vascular growth. The presence of *VEGF pathway* mutations provides reasoning for the usage of anti-angiogenic therapies. Tyrosine kinase inhibitors (TKIs) such as sorafenib, sunitinib, and pazopanib have been tested, though clinical benefit has been modest [27,40]. Interestingly, since *KDR* mutations can lead to independent activation, treatment with bevacizumab, which targets the *VEGF* ligand, may not be as effective [27]. These findings suggest that treatments targeting certain biomarkers, such as the presence of mutations in *KDR* or *FLT4*, may improve the efficacy of targeted therapy.

In our dataset, *PIK3CA* mutations were present in 38 of 359 samples (10.6%); these rates are similar to a prior analysis that reported a rate of approximately 8.0% (53/665) across angiosarcomas [27]. Activating mutations in *PIK3CA*, which encodes the *p110α* catalytic subunit of *PI3K*, are some of the most common across a variety of cancers [41]. The *p110α* encodes many key cell functions, such as growth, metabolism, and survival [27]. These mutations result in constitutive pathway signaling, causing uncontrolled tumor proliferation. The presence of *PIK3CA* mutations highlights a potential role for targeted therapy, with agents such as the *PI3Kα* inhibitor alpelisib and the pan-*PI3K* inhibitor copanlisib already being used for other cancers [42]. While reviews and genomic studies of angiosarcoma emphasize the role that biomarker-driven approaches can play, current clinical data are limited. This leaves room for further testing to determine the efficacy of such treatments in angiosarcomas.

In our analysis, we found significant co-occurrence among several gene pairs, including *FAT1* and *NOTCH2* (*p* < 0.001), *TP53* and *ATRX* (*p* < 0.001), *ARID1A* and *NOTCH1* (*p* < 0.001), *TP53* and *FAT1* (*p* < 0.001), and *ARID1A* and *NOTCH2* (*p* = 0.002). There also existed significant mutual exclusivity among two gene pairs: *KDR*-*FLT4* and *KDR*-*ATRX*.

Prior studies have shown that *TP53* commonly co-occurs with other genes, including *ATRX*, in angiosarcoma [27]. *TP53* serves as a tumor suppressor gene, controlling the cell cycle and apoptosis, while *ATRX* encodes for a chromatin-remodeling protein involved in telomere regulation and commonly exhibits loss of function in hepatic angiosarcoma [27,33]. Though limited data exists about the mechanism for this co-occurrence in angiosarcoma, Gulve et al. describe a potential mechanism for *TP53* and *ATRX* mutation co-occurrence. When *ATRX* is dysregulated, telomeres elongate. If *TP53* is also mutated, apoptosis and DNA repair are dysregulated. Gulve et al. hypothesize that changes in the structure of chromatin-remodeling proteins secondary to *ATRX* mutation prevent p53 proteins from binding and contributing to telomere elongation [43]. This derangement in normal DNA regulation likely contributes to the development of angiosarcoma.

In our analysis, we also found significant co-occurrence with *TP53* and *FAT1*. As previously discussed, *TP53* is a commonly mutated tumor suppressor gene. *FAT1* is also a commonly mutated gene in different cancers. Prior studies have shown that it regulates multiple pathways, including Wnt/β-catenin and *MAPK/ERK*, though its full role remains unclear. Mutations in the Wnt/β-catenin and *MAPK/ERK* pathways have been associated with epithelial–mesenchymal transition, a common process in tumorigenesis [44]. Wei et al. have found that multiple genes in these pathways are mutated in angiosarcoma [45]. Similarly, *TP53* mutations have been associated with epithelial–mesenchymal transition [46]. Co-mutation of *TP53-FAT1* may point to interactions between checkpoint failures (*TP53*) and issues with contact inhibition (*FAT1*). We hypothesize that this combination may lead to excess cell proliferation and growth, leading to increased mutational burden and an increased incidence of epithelial–mesenchymal transition [27,45].

Interestingly, although *PI3K* mutations are seen in primary breast angiosarcomas and *MYC* mutations are often seen in radiation-associated angiosarcomas, in our study, *PI3K* and *MYC* mutations did not exhibit statistically significant mutual exclusivity [27,30]. For instance, *PIK3CA* and *MYC* mutations showed mutual exclusivity with *p* = 0.142. Similarly, *KDR* mutations have been linked to breast angiosarcoma but did not demonstrate significant co-occurrence with *PI3K3CA* (*p* = 0.246) [27]. However, *MYC* and *KDR* alterations did exhibit significant mutual exclusivity (*p* = 0.022).

We also observed significant co-occurrence of *FAT1-NOTCH2*, *ARID1A-NOTCH*, and *ARID1A-NOTCH2*, suggesting potential interactions between *Notch* signaling, *FAT1* signaling (*Hippo*, Wnt), and *ARID1A* (chromatin remodeling) in angiosarcoma pathogenesis; these gene alterations have been identified in prior studies as well [15,27,34,45,47].

The co-alteration patterns identified in this cohort may carry therapeutic relevance based on their statistical significance and previously proposed therapeutic frameworks (Table 4). The co-mutation of *TP53-ATRX* (*p* < 0.001) has been associated with telomere maintenance and replication-stress vulnerability, suggesting possible sensitivity to *ATR* or *WEE1* inhibition [27,33,37,43]. Similarly, co-alterations involving *ARID1A-NOTCH1* (*p* < 0.001) and *ARID1A-NOTCH2* (*p* = 0.002) have been described in the context of chromatin-Notch-Hippo/Wnt pathway crosstalk, which may render tumors responsive to EZH2 or γ-secretase inhibitors [48,49]. The *FAT1-NOTCH2* interaction (*p* < 0.001) has also been linked to the chromatin-Notch-Hippo/Wnt pathway. The mutual exclusivity patterns observed, such as the VEGFR2 versus VEGFR3-driven angiogenic programs, have been proposed to inform the tailored use of VEGFR-targeted TKIs [27,40]. Finally, the co-alteration between *TP53-FAT1* (*p* < 0.001) further highlights the potential interactions between tumor suppressor and cell-adhesion/Hippo pathway regulators [27,45]. With further research and study, these signaling dependencies present in angiosarcomas may be leveraged for future therapeutic exploration.

Significant mutual exclusivity was less common than co-occurrence, only occurring in *KDR*-*FLT4* (*p* = 0.022) and *KDR-ATRX* (*p* = 0.028). These findings are consistent with prior studies reporting *KDR* and *FLT4* mutations in approximately 10–15% of angiosarcomas [15,27,38]. In our analysis, *KDR* and *FLT4* mutations were mutually exclusive (*p* = 0.022), which may suggest redundancy within the *VEGF* signaling pathway.

We also found *KDR-ATRX* to be mutually exclusive, which may indicate distinct biologies in angiosarcoma subclasses: angiogenesis dysfunction (*KDR*) and chromatin/telomere dysfunction (*ATRX*). While both of these mutations are present in angiosarcoma, this specific exclusivity has not been emphasized previously.

Overall, these patterns suggest that angiosarcomas can develop through different pathways, whether in tandem with other genes or separately. Recognizing these patterns can serve to assist in further biomarker research and treatment strategies.

Angiosarcoma has been shown to metastasize at higher rates than other soft tissue sarcomas [9]. The majority of samples in our study were taken from primary angiosarcoma sites (*n* = 252), though we did include metastatic samples (*n* = 79). There were several significant variations between primary and secondary tumor sites. Notably, *MAPK7* mutations solely occurred in primary tumors (*p* = 0.0253). This is surprising as prior studies of different primary cancers, including ovarian and bone, have shown an association between *MAPK7* mutations and metastasis. *MAPK7* is a driver of epithelial–mesenchymal transition, and its sole occurrence in primary tumors suggests that it might play a role in angiosarcoma progression and development. However, this was based on a small sample size, and further studies would be helpful to further understand *MAPK7*’s role in angiosarcoma [50,51].

Mutations in *ZFHX4* (*p* = 0.0392), *FGFR1* (*p* = 0.0436), *MSI2* (*p* = 0.0437), *HIST1H1C* (*p* = 0.0467), and *TOP1* (*p* = 0.0497) were significantly more common in metastatic tumors compared to primary tumors. *ZFHX4* has been associated with metastasis in ovarian cancer, hypothesized to be secondary to its ability to degrade the extracellular matrix [52]. It is possible that *ZFHX4* has a similar mechanism in angiosarcoma metastasis. Similarly, *MSI2* and *HIST1H1C* have been associated with metastasis of pancreatic cancer, while *TOP1* is associated with metastatic breast cancer [53,54,55]. *FGFR1* has been associated with suppression of metastasis in angiosarcoma, so loss-of-function mutations could be associated with metastasis [56].

## 5. Conclusions

This study has several important limitations. A primary constraint is the lack of clinical information in the AACR Project GENIE database, including prior cancer history and treatments such as radiotherapy, which is relevant given known associations with secondary angiosarcoma [7]. The database also does not specify whether samples were taken from primary, recurrent, or metastatic tumors, making it difficult to assess mutation patterns based on tumor origin. Additionally, the primary location of the tumor is generally unknown, except in clear cases involving the breast or liver. The dataset’s cross-sectional design and absence of follow-up data limit our ability to analyze trends in incidence, treatment response, or survival outcomes. Interpretation of racial differences is also challenged by missing demographic information, with approximately 18% of patients lacking race data. Lastly, the small sample size and potential variability in data collection across contributing centers may affect the consistency and generalizability of our findings.

Despite these limitations, our study contributes novel insights into the genomic and demographic landscape of angiosarcoma, offering perspectives not fully addressed in prior studies. By analyzing the AACR GENIE dataset, we identified recurrent alterations in pathways central to angiosarcoma, including *TP53*, *VEGF* (*KDR/FLT4*), and *PI3K* (*PIK3CA*), as well as tumor suppressors such as *ATM* and *ARID1A*, chromatin remodelers like *KMT2D*, and regulators of contact inhibition such as *FAT1*. Further analysis of subtypes displayed patterns such as *MYC* enrichment in breast angiosarcoma and *TP53* in liver angiosarcoma, as well as differences in incidence by sex and race. Co-occurrence of *TP53-ATRX* and *FAT1-NOTCH* suggests the presence of interacting sets of mutations, while mutual exclusivity of *KDR* and *FLT4* suggests distinct *VEGF*-driven angiogenic pathways. These findings reinforce the concept that angiosarcoma is not a single entity but rather a heterogeneous malignancy with a diverse molecular makeup. Future studies should aim to further understand the mechanisms of these mutations and to further elucidate their clinical significance. By highlighting this aspect of angiosarcoma, our work complements prior literature and offers additional insight into the molecular epidemiology of this rare malignancy.

## Figures and Tables

**Figure 1 cancers-17-03663-f001:**
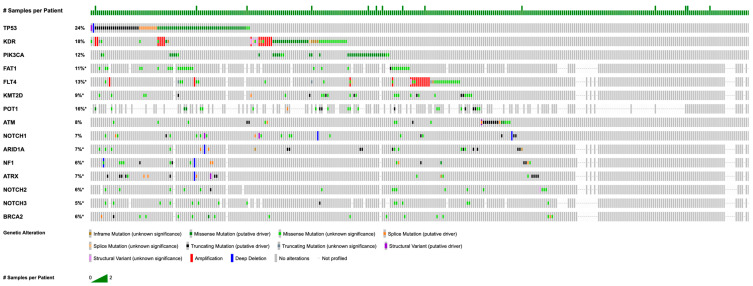
OncoPrint of common genetic alterations in angiosarcoma samples. Asterisk (*) indicates incomplete sample profiling.

**Figure 2 cancers-17-03663-f002:**
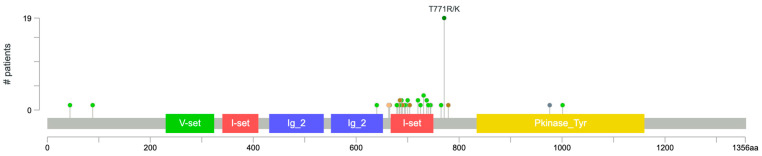
Common mutation locales in the *KDR* gene. Green dot: missense mutation. Brown dot: inframe mutation. Orange dot: splice mutation.

**Figure 3 cancers-17-03663-f003:**
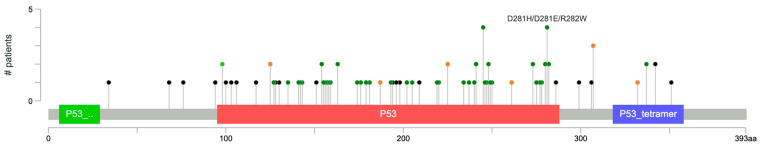
Common mutation locales in the *TP53* gene. Green dot: missense mutation. Black dot: truncating mutation. Orange dot: splice mutation.

**Table 1 cancers-17-03663-t001:** Demographic information of overall angiosarcoma patients, including both liver and breast angiosarcomas, compiled from the AACR GENIE database.

Demographics	Category	*n* (%)
Sex	Male	141 (40.8%)
	Female	194 (56.1%)
	Unknown	11 (3.2%)
Age category	Adult	355 (98.9%)
	Pediatric	4 (1.1%)
Ethnicity	Non-Hispanic	264 (76.3%)
	Unknown/Not Collected	51 (14.7%)
	Hispanic	31 (9.0%)
Race	Asian	24 (6.9%)
	White	239 (69.1%)
	Black	20 (5.8%)
	Other	28 (8.1%)
	Pacific Islander	1 (0.3%)
	Not collected	4 (1.2%)
	Unknown	30 (8.7%)
Sample Type	Primary	252 (70.2%)
	Metastasis	79 (22.0%)
	Not Collected	16 (4.5%)
	Unspecified	12 (3.3%)

**Table 2 cancers-17-03663-t002:** Statistically significant gene mutations in angiosarcoma patients by demographic category.

Gene (Chi-Squared)	Asian, *n* (%)	Non-Asian, *n* (%)	*p Value*
*CHEK1*	2 (8.33%	0 (0.00%)	7.950 × 10^−3^
*RBM10*	2 (8.33%)	0 (0.00%)	8.197 × 10^−3^
*CBL*	3 (12.50%)	3 (1.13%)	8.580 × 10^−3^
*EXT1*	1 (100%)	0 (0.00%)	0.0115
*FGF3*	2 (8.70%)	0 (0.00%)	0.0116
*BRAF*	4 (16.00%)	8 (2.95%)	0.0124
*SETD2*	3 (12.50%	5 (1.91%)	0.0221
**Gene (Chi-Squared)**	**White, *n* (%)**	**Non-White, *n* (%)**	** *p Value* **
*MYC*	4 (8.89%)	64 (26.56%)	0.0121
*GLI3*	0 (0.00%)	3 (13.64%)	0.0307
*MAPK7*	0 (0.00%)	1 (50.00%)	0.0417
**Gene (Chi-Squared)**	**Black, *n* (%)**	**Non-Black, *n* (%)**	** *p Value* **
*PRPF40B*	2 (100%)	1 (2.50%)	3.484 × 10^−3^
*CREBBP*	3 (15.00%)	3 (1.13%)	5.147 × 10^−3^
*AURKA*	2 (10.00%)	1 (0.38%)	0.0135
*KDM6A*	2 (10.00%)	2 (0.75%)	0.0258
*PARP1*	2 (14.29%)	2 (1.12%)	0.0271
*STAT3*	2 (10.00%)	2 (0.79%)	0.0279
*PPM1D*	2 (11.11%)	3 (1.36%)	0.0472
**Gene (Chi-Squared)**	**Pacific-Islander *n* (%)**	**Non-Pacific Islander, *n* (%)**	** *p Value* **
*PMS1*	1 (100%)	1 (0.35%)	7.042 × 10^−3^
*BORCS8-MEF2B*	1 (100%)	2 (0.68%)	0.0101
*FOXA1*	1 (100%)	3 (1.16%)	0.0154
*CYLD*	1 (100%)	3 (1.23%)	0.0163
*CALR*	1 (100%)	5 (2.02%)	0.0241
*PIK3CB*	1 (100%)	5 (2.55%)	0.030
*ARID1B*	1 (100%)	10 (3.57%)	0.0391
**Gene (Chi-Squared)**	**Male, *n* (%)**	**Female, *n* (%)**	** *p Value* **
*MYC*	8 (5.84%)	66 (34.02%)	2.14 × 10^−10^
*POT1*	20 (27.78%)	6 (6.52%)	3.870 × 10^−4^
*NTRK2*	1 (0.52%)	8 (5.93%)	4.232 × 10^−3^
*FGFR4*	0 (0.00%)	6 (4.35%)	4.578 × 10^−3^
*PIK3C2B*	0 (0.00%)	6 (12.50%)	8.582 × 10^−3^
*FAT1*	19 (14.84%)	10 (5.71%)	9.757 × 10^−3^
*HRAS*	14 (6.86%)	3 (2.08%)	0.0456

**Table 3 cancers-17-03663-t003:** Common gene mutations in angiosarcoma of the breast.

Gene	Number of Samples by Mutation	Total Breast Samples	Percentage
*MYC*	35	73	47.90%
*KDR*	17	73	23.30%
*FLT4*	11	73	15.10%
*PIK3CA*	9	73	12.30%
*KMT2D*	10	73	13.70%
*TP53*	4	73	5.50%

**Table 4 cancers-17-03663-t004:** Common co-alteration patterns and potential therapeutic targets.

Gene Pair	*p* Value	Impact of Mutation	Therapeutic Implications
*TP53-ATRX*	<0.001	Dysregulation of telomere maintenance and DNA repair	Sensitivity to CHK1/2 or WEE1 inhibition
*ARID1A-NOTCH1*	<0.001	Dysregulation of chromatin remodeling via chromatin–Notch–Hippo/Wnt crosstalk	Sensitivity to EZH2 or γ-secretase inhibitors
*ARID1A-NOTCH2*	0.002	Dysregulation of chromatin remodeling via chromatin–Notch–Hippo/Wnt crosstalk	Sensitivity to EZH2 or γ-secretase inhibitors
*KDR-FLT4*	0.022	Dysregulation of VEGFR-driven angiogenesis	Sensitivity to VEGFR-targeted TKIs
*TP53-FAT1*	<0.001	Dysregulation of telomere maintenance, DNA repair, epithelial–mesenchymal transition	Sensitivity to CHK1/2, WEE1, or Wnt/β-catenin inhibition

## Data Availability

The data was accessed and collected on 13 August 2025, from the cBioPortal (v18.0-public) software, which includes clinical and genomic data from 2017 onward.

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
