# Peer review of "Genomic and Demographic Characteristics of Angiosarcoma as Described in the AACR Project GENIE Registry"

_cancers, 2025, doi:10.3390/cancers17223663_

Round 1
Reviewer 1 Report
Comments and Suggestions for Authors
The data presented in this study are valuable in advancing our understanding of angiosarcoma biology, despite limitations related to the lack of clinical treatment information and limited annotation of tumor site of origin.
This work reinforces the concept that angiosarcoma biology is heterogeneous and varies according to the anatomic site of origin, suggesting that distinct initiating events and molecular pathways can converge to produce similar histologic phenotypes. The findings are consistent with prior reports based on smaller cohorts and contribute meaningfully to the existing literature.
Of note, citation 35 could be included among the references discussing differences in mutational profiles by primary site.
TMB analysis is described in the methods but no results presented about TMB in the cohort.
Expanding on breast angiosarcoma alterations in the context of previously published data could strengthen the manuscript. Including a table summarizing the cases and their specific genomic alterations could make this section more informative and easier to interpret.
Regarding PI3K alterations, these have been associated with primary breast angiosarcoma. Did you observe any pattern of mutual exclusivity with MYC amplification, which is typically seen in radiation-associated angiosarcomas? Similarly, you mention KDR mutations being linked to primary breast angiosarcoma; could you clarify whether KDR alterations co-occurred with PI3K mutations or MYC amplification?
Additionally, did FLT4 alterations co-exist with MYC amplification in your cohort? You note prior reports of this association in radiation-related breast angiosarcoma; clarification of whether the four MYC-amplified cases in your dataset showed concurrent FLT4 alterations would be helpful.
The observed patterns of co-alteration (e.g., TP53–ATRX, ARID1A–NOTCH1/2, FAT1–NOTCH2) and mutual exclusivity (KDR–FLT4, KDR–ATRX) provide a strong framework for inferring treatment susceptibility, the manuscript could link these to potential therapeutic implications. For instance, TP53–ATRX co-mutation may identify tumors with telomere maintenance and replication-stress vulnerability, suggesting possible sensitivity to ATR or WEE1 inhibition. ARID1A–NOTCH and FAT1–NOTCH interactions point toward chromatin–Notch–Hippo/Wnt crosstalk, which could render tumors responsive to EZH2, ATR, or γ-secretase inhibitors, and potentially Wnt/YAP pathway modulators. KDR–FLT4 mutual exclusivity implies distinct VEGFR2- versus VEGFR3-drivenangiogenic programs, which could inform the tailored use of VEGFR-targeted TKIs. Integrating these insights, perhaps as a concise table summarizing recurrent co-alterations and their plausible therapeutic vulnerabilities—would enhance the translational impact of the study.
Overall, this dataset provides strong foundational insight into recurrent and co-occurring genetic alterations in angiosarcoma, offering a valuable platform for the development of future molecularly targeted therapeutic strategies.
Lastly, please correct the reference formatting around line 81, where bracket placement appears inconsistent.
Author Response
Comments 1: The data presented in this study are valuable in advancing our understanding of angiosarcoma biology, despite limitations related to the lack of clinical treatment information and limited annotation of tumor site of origin. This work reinforces the concept that angiosarcoma biology is heterogeneous and varies according to the anatomic site of origin, suggesting that distinct initiating events and molecular pathways can converge to produce similar histologic phenotypes. The findings are consistent with prior reports based on smaller cohorts and contribute meaningfully to the existing literature. Overall, this dataset provides strong foundational insight into recurrent and co-occurring genetic alterations in angiosarcoma, offering a valuable platform for the development of future molecularly targeted therapeutic strategies. Of note, citation 35 could be included among the references discussing differences in mutational profiles by primary site.
Response 1: Thank you for pointing this out. We agree with this comment. Therefore, we have included this citation as a reference. Notably, since we have chosen to use this reference earlier in the text, it is now reference 15 rather than 35. This source is relevant to discussions of primary tumor site in the following lines: page 3, paragraph 1, line 81. The revisions are in red text.
Comments 2: TMB analysis is described in the methods but no results presented about TMB in the cohort.
Response 2: Thank you for pointing this out. We agree with this comment. This sentence in the methods section was included in error, and we did not assess TMB. Therefore, we have removed the sentence describing how tumor mutational burden would be assessed. Please see page 4, paragraph 4, line 144.
Comments 3: Expanding on breast angiosarcoma alterations in the context of previously published data could strengthen the manuscript. Including a table summarizing the cases and their specific genomic alterations could make this section more informative and easier to interpret.
Response 3: Thank you for pointing this out. We agree with this comment. Therefore, we have added a paragraph summarizing mutations seen among breast adenocarcinoma and additionally included a table showing percentage of samples with each mutation. We hope this change makes our data easier to interpret. Please see page 12, paragraph 5, lines 411-423 and additionally please see Table 3 on page 13, line 424. The revisions are in red text. "To better contextualize the breast angiosarcoma subset in our cohort, we compared the alterations identified in the GENIE data with those reported in prior genomic series. Among the 73 breast angiosarcoma samples (representing 72 patients), we found recurrent MYC amplification (35/73, 47.9%), as well as alterations in KDR/VEGFR2 (17/73, 23.3%), FLT4/VEGFR3 (11/73, 15.1%), PIK3CA (9/73, 12.3%), KMT2D (10/73, 13.7%), and TP53 (4/73, 5.5%) (Table 3). These trends are in line with previously described differences between primary and secondary breast angiosarcoma, in which PI3K pathway and chromatin-modifying gene alterations (such as PIK3CA and KMT2D) are more common in primary tumors [15,27]. MYC amplification, often accompanied by FLT4, has been repeatedly documented in radiation-associated disease [27,30]. The presence of KDR and FLT4, both VEGF-pathway alterations, within this subgroup underscores the angiogenic signaling dependence that has been described in earlier studies. This may inform future exploration of biomarker-driven anti-angiogenic strategies [27]."
Comments 4: Regarding PI3K alterations, these have been associated with primary breast angiosarcoma. Did you observe any pattern of mutual exclusivity with MYC amplification, which is typically seen in radiation-associated angiosarcomas? Similarly, you mention KDR mutations being linked to primary breast angiosarcoma; could you clarify whether KDR alterations co-occurred with PI3K mutations or MYC amplification?
Response 4: Thank you for pointing this out. We agree with this comment. Therefore, we have included analysis of PI3K/MYC, KDR/PI3K, and MYC/KDR co-occurrence. We have discussed that while co-occurrence might have been expected based upon prior studies, we did not find significant rates of co-occurrence with the exception of MYC/KDR. Please see page 16, paragraph 3, lines 568-574 for the appropriate changes in the manuscript. The revisions are in red text. "Interestingly, although PI3K mutations are seen in primary breast angiosarcomas and MYC mutations are often seen in radiation-associated angiosarcomas, in our study PI3K and MYC mutations did not exhibit statistically significant mutual exclusivity [27,30]. For instance, PIK3CA and MYC mutations showed mutual exclusivity with p=0.142. Similarly, KDR mutations have been linked to breast angiosarcoma but did not demonstrate significant co-occurrence with PI3K3CA (p=0.246) [27]. However MYC and KDR alterations did exhibit significant mutual exclusivity (p=0.022)."
Comments 5: Additionally, did FLT4 alterations co-exist with MYC amplification in your cohort? You note prior reports of this association in radiation-related breast angiosarcoma; clarification of whether the four MYC-amplified cases in your dataset showed concurrent FLT4 alterations would be helpful.
Response 5: Thank you for pointing this out. We agree with this comment. Therefore, we have included analysis of FLT4/MYC co-occurrence. We have discussed that co-occurrence might have been expected based upon prior studies and that we did find significant rates of co-occurrence. Please see page 12, paragraph 2, lines 388-389 for the appropriate changes in the manuscript. The revisions are in red text. "In our study, FLT4 mutations were prevalent among breast angiosarcomas (n=11, 15.1%) and did co-occur with MYC amplifications (p<0.001), consistent with the theory that many breast angiosarcomas develop secondary to radiation for primary breast cancer."
Comments 6: The observed patterns of co-alteration (e.g., TP53–ATRX, ARID1A–NOTCH1/2, FAT1–NOTCH2) and mutual exclusivity (KDR–FLT4, KDR–ATRX) provide a strong framework for inferring treatment susceptibility, the manuscript could link these to potential therapeutic implications. For instance, TP53–ATRX co-mutation may identify tumors with telomere maintenance and replication-stress vulnerability, suggesting possible sensitivity to ATR or WEE1 inhibition. ARID1A–NOTCH and FAT1–NOTCH interactions point toward chromatin–Notch–Hippo/Wnt crosstalk, which could render tumors responsive to EZH2, ATR, or γ-secretase inhibitors, and potentially Wnt/YAP pathway modulators. KDR–FLT4 mutual exclusivity implies distinct VEGFR2- versus VEGFR3-driven angiogenic programs, which could inform the tailored use of VEGFR-targeted TKIs. Integrating these insights, perhaps as a concise table summarizing recurrent co-alterations and their plausible therapeutic vulnerabilities—would enhance the translational impact of the study.
Response 6: Thank you for pointing this out. We agree with this comment. Therefore, we have included an additional paragraph discussing some of the therapeutic implications and also included a summary table summarizing major co-mutations and therapeutic implications. Please see page 16, paragraph 5, lines 581-595 and Table 4 on page 17, line 596. The revisions are in red text. "The co-alteration patterns identified in this cohort may carry therapeutic relevance based on their statistical significance and previously proposed therapeutic frameworks (Table 4). The co-mutation of TP53-ATRX (p<0.001) has been associated with telomere maintenance and replication-stress vulnerability, suggesting possible sensitivity to ATR or WEE1 inhibition [27,33,37,43]. Similarly, co-alterations involving ARID1A-NOTCH1 (p<0.001) and ARID1A-NOTCH2 (p=0.002) have been described in the context of chromatin-Notch-Hippo/Wnt pathway crosstalk, which may render tumors responsive to EZH2 or γ-secretase inhibitors [48,49]. The FAT1-NOTCH2 interaction (p<0.001) has also been linked to the chromatin-Notch-Hippo/Wnt pathway. The mutual exclusivity patterns observed, such as the VEGFR2 versus VEGFR3-driven angiogenic programs, have been proposed to inform the tailored use of VEGFR-targeted TKIs [27,40]. Finally, the co-alteration between TP53-FAT1 (p<0.001) further highlights the potential interactions between tumor suppressor and cell-adhesion/Hippo pathway regulators [27,45]. With further research and study, these signaling dependencies present in angiosarcomas may be leveraged for future therapeutic exploration."
Comments 8: Lastly, please correct the reference formatting around line 81, where bracket placement appears inconsistent.
Response 8: Thank you for pointing this out. We have fixed the brackets around our references on page 3, paragraph 1, line 86. The revisions are in red text.
Reviewer 2 Report
Comments and Suggestions for Authors
This manuscript provides a valuable summary of mutation data on angiosarcoma from the AACR database, representing a highly useful contribution to the field. I have only one comment. 
Cutaneous angiosarcomas are generally categorized into three subtypes: scalp angiosarcoma, radiation-associated angiosarcoma, and lymphedema-associated angiosarcoma. The latter two are characterized by MYC amplification as a distinctive genetic alteration.
In the present study, gene alterations according to tumor site do not appear to be clearly described. If site-specific genetic alteration data are available, including them would further enhance the significance of this work. I encourage the authors to consider this point.
Author Response
Comments 1: This manuscript provides a valuable summary of mutation data on angiosarcoma from the AACR database, representing a highly useful contribution to the field. I have only one comment. Cutaneous angiosarcomas are generally categorized into three subtypes: scalp angiosarcoma, radiation-associated angiosarcoma, and lymphedema-associated angiosarcoma. The latter two are characterized by MYC amplification as a distinctive genetic alteration.In the present study, gene alterations according to tumor site do not appear to be clearly described. If site-specific genetic alteration data are available, including them would further enhance the significance of this work. I encourage the authors to consider this point.
Response 2: Thank you for pointing this out. Although we agree this would be an interesting datapoint, unfortunately, version 18 of the cBio AACR GENIE database does not specify the site of collection for samples. Notably, the database does differentiate between general angiosarcoma and breast/liver angiosarcoma. We were therefore able to discuss mutations specific to angiosarcoma of the breast and liver. We do acknowledge this unfortunate limitation to our study in page 18, paragraph 6, lines 637-640.
Round 2
Reviewer 2 Report
Comments and Suggestions for Authors
No comments.